# Impact of health information technology optimization on clinical quality performance in health centers: A national cross-sectional study

**Robert Baillieu**[1], **Hank Hoang**[2‡], **Alek Sripipatana**[2‡], **Suma Nair**[2], **Sue C. Lin**[2]*

**1** Robert Graham Center for Policy Studies in Primary Care Washington, Washington, DC, United States of America, **2** US Department of Health and Human Service, Health Resources and Services Administration, Bureau of Primary Health Care, Office of Quality Improvement, Rockville, MD, United States of America

☯ These authors contributed equally to this work.
‡ These authors also contributed equally to this work.
* slin@hrsa.gov

**Data Availability Statement:** The Health Resources and Services Administration/Bureau of Primary Health Care (HRSA/BPHC)'s Health Center

## Abstract

### Background

Delivery of preventive care and chronic disease management are key components of a high functioning primary care practice. Health Centers (HCs) funded by the Health Resources and Services Administration (HRSA) have been delivering affordable and accessible primary health care to patients in underserved communities for over fifty years. This study examines the association between health center organization's health information technology (IT) optimization and clinical quality performance.

### Methods and findings

Using 2016 Uniform Data System (UDS) data, we performed bivariate and multivariate analyses to study the association of Meaningful Use (MU) attestation as a proxy for health IT optimization, patient centered medical home (PCMH) recognition status, and practice size on performance of twelve electronically specified clinical quality measures (eCQMs). Bivariate analysis demonstrated performance of eleven out of the twelve preventive and chronic care eCQMs was higher among HCs attesting to MU Stage 2 or above. Multivariate analysis demonstrated that Stage 2 MU or above, PCMH status, and larger practice size were positively associated with performance on cancer screening, smoking cessation counseling and pediatric weight assessment and counseling eCQMs.

### Conclusions

Organizational advancement in MU stages has led to improved quality of care that augments HCs patient care capacity for disease prevention, health promotion, and chronic care management. However, rapid technological advancement in health care acts as a potential source of disparity, as considerable resources needed to optimize the electronic health

Program makes the Uniform Data Systems (UDS) available, for free, to the public in an electronic format. This action is being taken pursuant to the Freedom of Information Act (5 U.S.C. 552(a)(2) (D)) (FOIA). URL: https://www.hrsa.gov/foia/UDS-public-use.html

**Funding:** This paper was submitted as part of the Health Resources and Services Administration (HRSA) Collection, which is sponsored and funded by HRSA.

**Competing interests:** The authors have declared that no competing interests exist.

record (EHR) and to undertake PCMH transformation are found more commonly among larger HCs practices. Smaller practices may lack the financial, human and educational assets to implement and to maintain EHR technology. Accordingly, targeted approaches to support small HCs practices in leveraging economies of scale for health IT optimization, clinical decision support, and clinical workflow enhancements are critical for practices to thrive in the dynamic value-based payment environment.

## Introduction

In 2011, the Centers for Medicare & Medicaid Services (CMS) implemented the Medicare and Medicaid Electronic Health Record (EHR) Incentive Programs to encourage clinicians and practices to adopt, implement, upgrade, and demonstrate meaningful use of certified electronic health record technology (CEHRT) [1, 2]. The CMS Meaningful Use (MU) program, which is now known as Promoting Interoperability, continued to compel all clinical practices to optimize EHR functionality in order to achieve integrated platforms for better patient care, lower healthcare costs, and to promote patient engagement. In particular, Stage 2 of MU sought to expand the adoption of advanced EHR functions that included: clinical decision support; electronic prescribing; health information exchange; patient-tailored health and disease management tools; secure electronic communication between patients and providers; health education materials; and monitoring of clinical quality metrics [3, 4]. Other EHR functions, such as the presence of clinical reminders and the production of clinic-level data, are common to all CEHRT [1].

The Institute for Healthcare Improvement (IHI) first promulgated the Triple Aim in 2008, as a means of improving the experience of care, the health of populations, and reducing per capita costs of health care. Since that time, value-based care delivery programs have been guided by national clinical quality metrics and the principles of continuous quality improvement [5, 6]. Underlying this is the EHR, as it provides systematic assistance to clinical activities, communication, clinical decision support and enhanced reporting and monitoring of quality metrics. In 2015, 86% of office-based physicians in the United States had adopted an EHR, but the sophistication of EHR technology and its capacity for quality reporting varied across practice settings [7–10]. Advancing the uptake of supports provided by EHR and health information technology (health IT) innovations has the potential to promote parity in health IT implementation across practice settings [11].

Studies have demonstrated a positive association between MU implementation and improved process quality metrics in preventive screening, diabetes control, maternal and child health measures in primary care clinics that leveraged health IT for patient engagement and care coordination [12, 13]. Embedded decision support tools such as electronic reminders have significant impact on the uptake of preventative care and performance of pertinent preventative risk assessments in the clinical setting [14, 15]. In many practices, health IT has been used to disseminate evidence-based care guidelines and to provide clinical alerts that enhance patient safety and decrease mortality [16–19].

In 2016, the Health Resources and Services Administration's (HRSA) Health Center Program (HCP) was comprised of over 1,300 health centers (HCs) operating more than 11,000 primary care service delivery sites [20]. These HCs provided comprehensive, affordable and quality primary health care to nearly 26 million individuals in every U.S. state, the District of Columbia, Puerto Rico, the U.S. Virgin Islands, and the Pacific Basin. From 2011 to 2016,

EHR adoption rose from 65% to 95% among HCs. To standardize the monitoring of clinical quality performance, HRSA collects data on electronically specified clinical quality measures (eCQMs) annually; twelve eCQMs were collected in 2016. Additionally, 66% of HCs achieved primary care patient-centered medical home (PCMH) recognition by meeting national standards for primary care that emphasized care coordination, comprehensive care and on-going clinical quality improvement that included leveraging health IT [20]. The specific aim of this study is to examine the association of Stage 2 MU attainment on eCQM performance among HCs, and to explore surrogate levers of health IT implementation, specifically PCMH recognition and practice size, in clinical quality performance.

## Methods

### Data acquisition

Data came from HRSA's 2016 Uniform Data System (UDS), which is an administrative data set containing information on patient sociodemographic characteristics, primary care services provision, healthcare workforce, clinical quality measures (CQMs), and Meaningful Use (MU) attestation that are reported and aggregated at the health center organizational level. MU attestation for Stage 2 or above was identified through HCs self-report to the following two questions: 1) Are your eligible providers participating in the CMS EHR Incentive Program commonly known as "Meaningful Use"?; and 2) If yes, at what stage of Meaningful Use are the majority of your participating providers who have most recently received incentive payments? In addition, PCMH recognition status was ascertained for HCs who received certification from national organizations and state-based initiatives. Finally, the study categorized practice size by number of physician full time equivalents (FTEs) in HCs as follows: 1) small practice was defined as HCs with 0–5 physician FTEs; 2) medium practice was defined as 6–20 physician FTEs; and 3) large practice was defined as 21 or more physician FTEs [21, 22].

### Study design

A descriptive cross-sectional study of the association of health IT optimization, PCMH recognition status, and practice size on clinical quality measure performance was conducted. It was hypothesized that HCs that are larger and have achieved MU stage 2 or above, as well as PCMH recognition, would demonstrate better clinical quality performance. The dependent variables of interest were the twelve preventive and chronic care eCQMs reported in the 2016 UDS. The eight preventive measures consisted of the following: 1) cervical cancer screening; 2) colorectal cancer screening; 3) adult body mass index (BMI) screening and follow-up plan; 4) weight assessment and counseling for nutrition and physical activity for children and adolescents; 5) tobacco use screening and cessation intervention; 6) depression screening and follow-up plan; 7) childhood immunization; and 8) dental sealant for children between 6–9 years old. The four chronic care measures were as follows: 1) aspirin therapy for patients with ischemic vascular disease; 2) blood pressure control (as defined by hypertensive patients with a blood pressure less than 140/90mmHg); 3) uncontrolled diabetes (i.e. diabetic patients with an HbA1c > 9%); and 4) asthma pharmacologic therapy [20].

### Data analysis

In Table 1, we conducted bivariate analysis to compare the mean percentages of patient sociodemographic attributes by practice size using an F test, as well as the HCs' attainment of PCMH recognition and MU Stage 2 or above using chi-square analysis. In Table 2, we

**Table 1. Characteristics of Health Centers (HC) and patients served by practice size.**

| Characteristics | Small Practice N = 685 | | Medium Practice N = 517 | | Large Practices N = 143 | | p-value |
|---|---|---|---|---|---|---|---|
| | Mean | SD | Mean | SD | Mean | SD | |
| **Race/ethnicity** | | | | | | | |
| Hispanic | 21.3% | 0.24 | 30.1% | 0.28 | 45.2% | 0.28 | **<0.01** |
| Non-Hispanic White | 52.5% | 0.52 | 42.6% | 0.31 | 33.1% | 0.27 | **<0.01** |
| Non-Hispanic Black | 20.5% | 0.26 | 25.0% | 0.27 | 18.7% | 0.21 | **<0.01** |
| Other | 9.3% | 0.19 | 7.0% | 0.14 | 8.8% | 0.17 | 0.06 |
| **Language Preferred** | | | | | | | |
| Patients Best Served in a language other than English | 13.9% | 0.20 | 21.8% | 0.24 | 31.6% | 0.24 | **<0.01** |
| **Age** | | | | | | | |
| 0–17 Years | 23.2% | 0.13 | 29.7% | 0.12 | 33.1% | 0.10 | **<0.01** |
| 18–64 Years | 67.0% | 0.13 | 61.4% | 0.11 | 58.9% | 0.08 | **<0.01** |
| 65 Years and Older | 10.0% | 0.07 | 8.9% | 0.05 | 7.9% | 0.04 | **<0.01** |
| **Household poverty level** | | | | | | | |
| ≤100% | 46.5% | 0.24 | 49.2% | 0.23 | 52.2% | 0.23 | **0.02** |
| 101–200% | 16.6% | 0.11 | 15.9% | 0.11 | 14.6% | 0.08 | 0.09 |
| >200% | 6.9% | 0.09 | 5.9% | 0.07 | 5.8% | 0.07 | 0.06 |
| Not reported | 30.2% | 0.27 | 29.3% | 0.26 | 27.5% | 0.26 | 0.50 |
| **Insurance status** | | | | | | | |
| Uninsured | 28.9% | 0.20 | 22.8% | 0.16 | 19.8% | 0.12 | **<0.01** |
| Medicaid/CHIP | 39.2% | 0.20 | 47.9% | 0.18 | 54.9% | 0.15 | **<0.01** |
| Medicare | 11.0% | 0.08 | 10.0% | 0.06 | 8.4% | 0.05 | **<0.01** |
| Private Insurance | 20.7% | 0.14 | 18.9% | 0.12 | 17.0% | 0.11 | **<0.01** |
| | | | | | | | Chi-Square P-value |
| **Patient-Centered Medical Home Recognition** | 52.6% | | 78.9% | | 93.7% | | **<0.01** |
| **Meaningful Use Stage 2 or 3 Attestation** | 38.3% | | 54.9% | | 72.0% | | **<0.01** |

Source: 2016 UDS Data

Bold numbers indicate p-value ≤ 0.05

examined the unadjusted association of clinical quality measure performance with MU attestation at Stage 2 or above in order to compare the means of eCQM by MU attestation stage using an F test. In Table 3, we looked at the association of practice size with eCQM performance using an F test statistic. Mean percentages have been reported in Tables 1–3 as it is the most commonly utilized measure of central tendency [23]. Finally, we carried out multiple linear regression analyses to assess the association of MU stage, PCMH status, and practice size on clinical quality performance. PCMH status was included due to previous established associations with clinical quality improvement in health center research [24–27]. After conducting the interaction testing, the interacting terms were found to be not statistically significant and thus were not retained in the regression model [28]. Clinical quality measures with statistically non-significant associations with MU stage and practice size from the bivariate analyses were not included in the multiple linear regression analyses. Control variables in the regression model included the following patient characteristics: percentage of racial/ethnic minority patients, percentage of patients at or below 100% of the federal poverty level (FPL), and percentage of uninsured patients. All statistical analyses were performed using SAS version 9.4.

**Table 2. Electronic Clinical Quality Measure (e-CQMs) performance by Meaningful Use (MU) attestation.**

| e-CQMs | MU Stage ≤ 1 | MU Stage ≥ 2 | |
|---|---|---|---|
| (mean percentages) | | | p-value |
| 1. Cervical Cancer Screening | 46.4% | 52.1% | **<0.01** |
| 2. Colorectal Cancer Screening | 34.1% | 40.0% | **<0.01** |
| 3. Adult Body Mass Index (BMI) Screening and Follow-Up Plan | 57.7% | 62.8% | **<0.01** |
| 4. Weight Assessment & Counseling for Nutrition & Physical Activity (PA) for Children & Adolescents | 52.6% | 60.6% | **<0.01** |
| 5. Diabetes A1C Poor Control | 35.0% | 31.6% | **<0.01** |
| 6. Ischemic Vascular Disease (IVD): Use of Aspirin or Another Antithrombotic | 76.1% | 78.2% | **0.01** |
| 7. Controlling High Blood Pressure | 60.8% | 62.7% | **<0.01** |
| 8. Tobacco Use: Screening and Cessation Intervention | 81.2% | 84.6% | **<0.01** |
| 9. Asthma Pharmacologic Therapy | 84.2% | 86.5% | **<0.01** |
| 10. Childhood Immunizations | 35.9% | 39.3% | **0.02** |
| 11. Depression Screening and Follow-Up Plan | 58.0% | 61.0% | **0.03** |
| 12. Dental Sealant for Children between 6–9 years | 48.4% | 47.3% | 0.49 |

Source: 2016 UDS Data

Bold numbers indicate p-value ≤ 0.05

## Results

Table 1 describes demographic characteristics of HCs at the organizational level and by practice size. When compared to smaller practices, large practices cared for: a higher mean percentage of Hispanic patients (45.2% versus 21.3%, P<0.01); patients who are best served in a language other than English (31.6% versus 13.9%, P<0.01); patients with a household poverty level at or below 100% FPL (52.2% versus 46.5%, P<0.02); and patients with Medicaid or Children's Health Insurance Program (CHIP) insurance (54.9% versus 39.2%, P<0.01). In contrast, small practices reported a higher mean percentage of Non-Hispanic White patients (52.5% versus 33.1%, P<0.01) and patients 65 years or older (10.0% versus 7.9%, P<0.01) in comparison to their larger counterparts. Moreover, small practices served higher percentages

**Table 3. Electronic Clinical Quality Measure (e-CQMs) performance by health center practice size.**

| e-CQMs | Small Practice | Medium Practice | Large Practice | |
|---|---|---|---|---|
| (mean percentages) | | | | p-value |
| 1. Cervical Cancer Screening | 44.5% | 53.6% | 57.7% | **<0.01** |
| 2. Colorectal Cancer Screening | 33.2% | 40.9% | 42.2% | **<0.01** |
| 3. Adult Body Mass Index (BMI) Screening and Follow-Up Plan | 58.9% | 62.0% | 62.4% | **0.03** |
| 4. Weight Assessment & Counseling for Nutrition & Physical Activity (PA) for Children & Adolescents | 52.0% | 61.3% | 63.9% | **<0.01** |
| 5. Diabetes A1C Poor Control | 34.6% | 32.4% | 30.0% | **<0.01** |
| 6. Ischemic Vascular Disease (IVD): Use of Aspirin or Another Antithrombotic | 75.6% | 78.5% | 80.0% | **<0.01** |
| 7. Controlling High Blood Pressure | 60.6% | 62.5% | 64.3% | **<0.01** |
| 8. Tobacco Use: Screening and Cessation Intervention | 80.4% | 85.0% | 86.6% | **<0.01** |
| 9. Asthma Pharmacologic Therapy | 83.8% | 86.3% | 89.2% | **<0.01** |
| 10. Childhood Immunizations | 33.2% | 40.9% | 45.8% | **<0.01** |
| 11. Depression Screening and Follow-Up Plan | 60.0% | 59.1% | 58.7% | 0.74 |
| 12. Dental Sealant for Children between 6–9 years | 48.9% | 46.7% | 48.8% | 0.41 |

Source: 2016 UDS Data

Bold numbers indicate p-value ≤ 0.05

of uninsured (28.9% versus 19.8%, P<0.01), Medicare (11.0% versus 8.4%, P<0.01), and privately insured patients (20.7% versus 17.0%, P<0.01) than large practices. The percentage of practices with PCMH recognition were as follows: 52.6% of small practices; 78.9% of medium practices; and 93.7% of large practices, (P<0.01). 38.3% of small practices attested to MU Stage 2 or above as compared with 54.9% of medium sized practices and 72.0% of large practices (P<0.01).

Table 2 contains eCQM performance by MU attestation status. Performance on eleven out of twelve eCQMs were significantly higher among HCs attesting to MU Stage 2 or above. Although the dental sealants for children eCQM, which was introduced in the 2015 UDS, demonstrated an inverse pattern, this finding was not statistically significant. Notably, significant differences in performance of 5 percentage points or more were observed in: cancer prevention measures of cervical cancer screening (46.4% vs. 52.1%); colorectal cancer screening (34.1% vs 40.0%); and obesity prevention measures for adult (57.7% vs. 62.8%) and pediatric patients (52.6% vs. 60.6%).

Table 3 presents the bivariate analysis of eCQMs performance by small, medium, and large practice size. Ten out of twelve eCQMs had the highest mean percentage among large HCs. In particular, we observed significant differences in performance of 5 or more percentage points when comparing eCQMs of small to large practices with respect to cervical cancer screening (44.5% vs. 57.7%), colorectal cancer screening (33.2% vs. 42.2%), obesity prevention measure for pediatric patients (52.0% vs 63.9%), tobacco use screening and cessation intervention (80.4% vs 86.6%), asthma pharmacologic therapy (83.8% vs 89.2%), and childhood immunization (33.2% vs. 45.8%). Similar to the findings in Table 2, the dental sealants for children eCQM showed a reverse pattern that was not statistically significant. In addition, the depression screening and follow-up plan eCQM had no statistical significance findings.

Table 4 describes results from multiple linear regressions performed between eCQMs as the dependent variables and MU stage, PCMH status, and HC practice size as the independent variables. MU Stage 2 or above was a significant predictor of performance on cancer prevention, obesity prevention, tobacco screening and cessation counseling, childhood immunization, and diabetes control measures. PCMH recognition was a significant predictor for all eCQMs except childhood immunization. With respect to practice size, large practice size was a significant positive predictor for cancer prevention, hypertension control, diabetes control, tobacco screening and cessation counseling, depression screening and follow-up plan, and childhood immunization measures. For prevention care eCQMs, MU Stage 2 or above, PCMH and practices size were significant positive predictors for colorectal cancer screening, cervical cancer screening, smoking cessation counseling, and pediatric weight assessment and counseling.

## Discussion

Our findings suggest that health IT optimization, PCMH transformation, and larger practice size correlate with better clinical quality performance in the majority of eCQMs reported by HRSA HCs. Health IT optimization by primary care practices facilitates quality improvement (QI) and enables effective implementation of PCMH to enhance care coordination, deliver high quality care, prevent unnecessary acute care visits and ultimately improve patient outcomes. It further holds the promise of better continuity of care, particularly for underserved populations that faces multiple competing priorities in accessing health care [29, 30]. Federal investments that accelerate health IT optimization in HCs through strengthening health IT infrastructure, as well as promoting targeted health IT training and technical assistance (T/TA) will continue to be critically important.

**Table 4. Multiple linear regression of Clinical Quality Measure (CQM) performance.**

| Elec-tronic CQM | Meaningful Use Stage 2 or Above | | | | | Patient Centered Medical Home | | | | | Practice Size Large (Ref = Small) | | | | | Practice Size Medium (Ref = Small) | | | | |
|---|---|---|---|---|---|---|---|---|---|---|---|---|---|---|---|---|---|---|---|---|
| | Coefficient | 95% CI | SE | t-value | p-value | Coefficient | 95% CI | SE | t-value | p-value | Coefficient | 95% CI | SE | t-value | p-value | Coefficient | 95% CI | SE | t-value | p-value |
| **Chronic Care CQM** | | | | | | | | | | | | | | | | | | | | |
| Ischemic Vascular Disease (IVD): Use of Aspirin or Another Antithrombotic | 1.07 | (-0.63, 2.77) | 0.87 | 1.24 | 0.22 | 3.18 | (1.28, 5.07) | 0.97 | 3.29 | <0.01 | 2.10 | (-0.92, 5.12) | 1.54 | 1.36 | 0.17 | 1.47 | (-0.45, 3.38) | 0.97 | 1.50 | 0.13 |
| Controlling High Blood Pressure | 1.11 | (-0.03, 2.24) | 0.58 | 1.91 | 0.06 | 1.81 | (0.55, 3.07) | 0.64 | 2.81 | 0.01 | 2.46 | (0.44, 4.48) | 1.03 | 2.39 | 0.02 | 0.75 | (-0.5,3 2.02) | 0.65 | 1.15 | 0.25 |
| Diabetes A1C Poor Control | -1.77 | (-3.18, -0.37) | 0.71 | -2.48 | 0.01 | -3.39 | (-4.95, -1.83) | 0.79 | -4.27 | <0.01 | -3.13 | (-5.62, -0.63) | 1.27 | -2.46 | 0.01 | -1.06 | (-2.64, 0.51) | 0.80 | -1.32 | 0.19 |
| Asthma Pharmacologic Therapy | 1.32 | (-0.38, 3.03) | 0.87 | 1.52 | 0.13 | 1.97 | (0.07, 3.87) | 0.97 | 2.03 | 0.04 | 2.78 | (-0.24, 5.81) | 1.54 | 1.81 | 0.07 | 1.06 | (-0.86, 2.97) | 0.97 | 1.08 | 0.28 |
| **Preventive Care** | | | | | | | | | | | | | | | | | | | | |
| Cervical Cancer Screening | 3.67 | (1.83, 5.51) | 0.94 | 3.92 | <0.01 | 3.28 | (1.23, 5.32) | 1.04 | 3.15 | <0.01 | 8.51 | (5.25, 11.78) | 1.66 | 5.12 | <0.01 | 6.08 | (4.02, 8.14) | 1.05 | 5.78 | <0.01 |
| Colorectal Cancer Screening | 3.40 | (1.43, 5.38) | 1.01 | 3.38 | <0.01 | 5.17 | (2.97, 7.37) | 1.12 | 4.61 | <0.01 | 4.62 | (1.11, 8.14) | 1.79 | 2.58 | 0.01 | 4.75 | (2.53, 6.97) | 1.13 | 4.20 | <0.01 |
| Adult BMI Screening & F/Up Plan | 4.37 | (1.92, 6.81) | 1.25 | 3.50 | <0.01 | 3.84 | (1.12, 6.56) | 1.39 | 2.77 | 0.01 | -0.88 | (-5.23, 3.46) | 2.22 | -0.40 | 0.69 | 0.46 | (-2.28, 3.21) | 1.40 | 0.33 | 0.74 |
| Weight Assessment & Counseling for Nutrition & Physical Activity for Children & Adolescents | 5.39 | (2.55, 8.22) | 1.45 | 3.73 | <0.01 | 4.03 | (0.87, 7.19) | 1.61 | 2.50 | 0.01 | 5.56 | (0.53, 10.60) | 2.57 | 2.17 | 0.03 | 5.45 | (2.27, 8.64) | 1.62 | 3.36 | <0.01 |
| Tobacco Use: Screening & Cessation Intervention | 1.87 | (0.06, 3.68) | 0.92 | 2.03 | 0.04 | 3.69 | (1.68, 5.71) | 1.03 | 3.60 | <0.01 | 3.85 | (0.64, 7.07) | 1.64 | 2.35 | 0.02 | 3.11 | (1.08, 5.14) | 1.04 | 3.00 | <0.01 |
| Depression Screening and Follow-up Plan | 2.73 | (-0.05, 5.51) | 1.42 | 1.93 | 0.05 | 4.65 | (1.56, 7.74) | 1.58 | 2.95 | <0.01 | -5.58 | (-10.51, -0.64) | 2.52 | -2.22 | 0.03 | -3.46 | (-6.58, -0.34) | 1.59 | -2.18 | 0.03 |
| Childhood Immunizations | 3.04 | (0.21, 5.87) | 1.44 | 2.11 | 0.04 | -1.07 | (-4.24, 2.10) | 1.62 | -0.66 | 0.51 | 9.70 | (4.71, 14.69) | 2.54 | 3.81 | <0.01 | 6.19 | (3.02, 9.36) | 1.62 | 3.83 | <0.01 |

Source: 2016 UDS Data

Model control for % of Minority Patients, Patients of Poverty Level 100% and Below, and Uninsured Patients; CI = confidence interval; SE = standard error

In demonstrating that successful health IT optimization can support improvement in clinical quality outcome measures, this study is aligned with previous analyses of these factors that impact HC performance [24–27]. The associations of MU Stage 2 or above, PCMH, and large practice size with diabetes control are very positive news for chronic care management. Furthermore, the positive association of PCMH recognition with nine eCQMs demonstrates the importance of PCMH transformation in clinical quality performance among HCs. Finally, as health IT optimization reaches the stage where health, social, economic, behavioral, and environmental data can be fully integrated to customize care for the patient, primary care teams might be able to more comprehensively address social determinants of health within the PCMH [31, 32].

An emerging body of research suggests that advanced EHR technologies are potentially associated with improved information sharing, enhanced patient interaction with the EHR, and less burdensome quality reporting [17, 18, 33, 34]. This could be especially true when practices customize their EHRs to better reflect clinical workflows, patient desired as well as provider and care team preferences. The significant improvement in cancer screening rates and preventive care delivery in those HCs that attested to Stage 2 MU or higher, for example, suggests that health IT optimization may be of benefit in augmenting a clinical encounter through patient reminders and other readily available electronic educational resources that promote health equity. The literature also demonstrates cost and time savings after configuration of an EHR to facilitate data collection to automatically report quality metrics [34, 35].

Overall, HCs have made great strides in health IT implementation and optimization. With respect to practice size, 72% of large HCs and 55% of medium-sized HCs successfully attested to MU Stage 2 or above. This is compared to the 60% of all U.S. office-based physicians (MD/DO) who reported meaningful use of certified health IT to the CMS EHR Incentive Programs in 2016. The positive association between practice size and MU in adult and pediatric preventive care suggests that successful implementation of clinical care and workflow supported by health IT contributes to reducing the burden of preventable chronic disease [19, 36–38]. Previous research suggests that the relationship between eCQM performance and practice size is in part attributable to available human and financial capital [39, 40]. In comparison to their larger counterparts, small and medium sized practices are more likely to experience lower physician to patient ratios and shorter consultation times [39]. Moreover, smaller practices are predominantly found in areas of higher economic need [21]. This has been determined to be an independent marker of lower health outcomes, possibly due to the higher morbidity associated with those underserved communities [41]. In this way, physicians in smaller practices face time constraints that potentially make adherence to quality guidelines difficult, while also treating a patient population that eschews preventative care in favor of acute management [42–44]. In addition, our study showed that small HCs disproportionately serve uninsured patients, which may impact their ability to allocate significant resources towards health IT. This finding suggests strategically targeting small and medium sized practices for health IT T/TA support in attaining CMS Promoting Interoperability Program requirements.

## Limitations

The UDS is an administrative dataset reported by HRSA-funded HCs and aggregated at the HC organizational level. Although HC may operate several health care clinical sites, data in the UDS cannot be filtered by delivery site. While certain elements of the UDS (e.g., eCQMs) are automatically extracted from the EHR, other elements of the UDS are self-reported including Health Information Technology Capabilities and Staffing data. In addition, staffing is captured as full-time equivalents, and not the actual number of physicians/providers.

## Policy implications

Practices that report being at MU Stage 2 or above experience tangible benefits in coordination of communication, patient care and data management. Our findings suggest that optimization of health IT, PCMH transformation, and practice size are closely related to enhanced quality of care and health outcomes. Given this association, HRSA has strategically aligned CQM reporting with eCQMs, where possible. However, the potential benefits of health IT optimization are not being realized across all HCs, particularly among smaller practices. The underlying financial, training, staffing and opportunity costs associated with the implementation and maintenance of EHR technology may be potential sources of disparity for those small or medium practices without access to significant human or financial capital. It is critical that safety net providers remain current with advances in health IT adoption and utilization in order to maximize quality of care, ensure patient safety, reduce health disparities, improve care coordination and augment public health reporting. Better understanding of those EHR functions that are most relevant and useful to smaller HCs would help direct HRSA's technical assistance assets and resource allocation efforts. Such assessments need to be ongoing and multidimensional since advancement in health IT and EHR technology is rapid and also uneven across practices of different sizes. Overcoming this disparity is an important way to support patient care in underserved communities and to promote access to all health centers.

## Author Contributions

**Conceptualization:** Robert Baillieu, Hank Hoang, Alek Sripipatana, Suma Nair, Sue C. Lin.

**Data curation:** Robert Baillieu, Hank Hoang, Alek Sripipatana, Sue C. Lin.

**Formal analysis:** Robert Baillieu, Sue C. Lin.

**Investigation:** Robert Baillieu, Hank Hoang, Alek Sripipatana, Suma Nair, Sue C. Lin.

**Methodology:** Robert Baillieu, Hank Hoang, Alek Sripipatana, Sue C. Lin.

**Project administration:** Suma Nair, Sue C. Lin.

**Resources:** Suma Nair.

**Software:** Sue C. Lin.

**Supervision:** Suma Nair, Sue C. Lin.

**Validation:** Robert Baillieu, Sue C. Lin.

**Visualization:** Robert Baillieu, Hank Hoang, Alek Sripipatana, Suma Nair, Sue C. Lin.

**Writing – original draft:** Robert Baillieu, Hank Hoang, Alek Sripipatana, Sue C. Lin.

**Writing – review & editing:** Robert Baillieu, Hank Hoang, Alek Sripipatana, Suma Nair, Sue C. Lin.

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
