## [Decision Letter · Decision Letter 0]

29 Nov 2019

PONE-D-19-29335

Impact of Health Information Technology Optimization on Clinical Quality Performance in Health Centers: A National Cross-Sectional Study

PLOS ONE

Dear Dr. Lin,

Thank you for submitting your manuscript to PLOS ONE. After careful consideration, we feel that it has merit but does not fully meet PLOS ONE’s publication criteria as it currently stands. Therefore, we invite you to submit a revised version of the manuscript that addresses the points raised during the review process.

We would appreciate receiving your revised manuscript by Jan 03 2020 11:59PM. To enhance the reproducibility of your results, we recommend that if applicable you deposit your laboratory protocols in protocols.io, where a protocol can be assigned its own identifier (DOI) such that it can be cited independently in the future. For instructions see: http://journals.plos.org/plosone/s/submission-guidelines#loc-laboratory-protocols

We look forward to receiving your revised manuscript.

Kind regards,

Mustafa Ozkaynak

Academic Editor

PLOS ONE

**Journal Requirements:**

Additional Editor Comments:

The manuscript reports on an important study but needs major revisions.

In summary,

- Statistical analysis has important concerns. Please rewrite this section using the feedback from reviewers.

- Introduction section can be enriched adding some relevant literature.

- Discussion section should be reorganized and enriched using the findings in the results section.

Please see reviewer comments for further details.

Reviewers' comments:

Reviewer's Responses to Questions

**Comments to the Author**

1. Is the manuscript technically sound, and do the data support the conclusions?

Reviewer #1: Partly

Reviewer #2: Yes

Reviewer #3: Yes

2. Has the statistical analysis been performed appropriately and rigorously? 

Reviewer #1: No

Reviewer #2: Yes

Reviewer #3: Yes

3. Have the authors made all data underlying the findings in their manuscript fully available?

Reviewer #1: Yes

Reviewer #2: Yes

Reviewer #3: Yes

4. Is the manuscript presented in an intelligible fashion and written in standard English?

Reviewer #1: Yes

Reviewer #2: Yes

Reviewer #3: Yes

5. Review Comments to the Author

Reviewer #1: For the statistical analysis seems incomplete. I am assuming you are aggregating the data at the clinic level, which means you are going to have for any outcome a range from 0-100% for a clinic meeting that quality measure, correct? if this is true then you are not doing bivariate (logistic) regression - you are doing linear regression (or you can do a probit model taking into account the N for each clinic).

or if you are modeling at the patient level then you should be doing a mixed model with a random effect at the clinic level.

Concerns regarding model selection - removing variables that do not meet significance (which I am assuming you are using cutoff of 0.05) is not an approach that is recommended for model selection. For your sample size you are not hurting on power so why not adjust by all factors of interest if you do not want to do true model selection?

Also, on any regression model you are making assumptions and you do not state you verify them in the statistical section.

For table 2, Only MU is addressed but you are interested in association across 3 key variables of which PCMH is not addresses in terms of describing outcomes (practice size in table 1).

Test statistics in table 1 and 2 were not mentioned and should be described in statistical section (what is p-value testing??).

The term Multivariate is not used appropriately, you are actually doing Multivariable linear regression.

For presenting modeling results you show coefficient and p-value, the results should show how much variation there is about the estimate.

Reviewer #2: This is an important topic given that IT optimization ( via meaningful use stage 2) is critical or meant to be critical to improve quality of care. This paper explores association of these measures using public data called UDS. Please see my detailed comments section by section below

Introduction

1. The paper needs to add findings form studies which reported on HIT implementation ( EHR adoption) and their impact of clinical quality outcomes. There is big literature on that which is missing in the introduction

2. There are other studies used the same data set published results for similar or different measures. Some of these studies should be reported

Methods

1. The method section should be formatted well with sub headings such as study design, data Acquisition, data analysis, validity etc.

2. They need to add hypothesis and state if hypothesis are supported or not.

Results

It is straightforward and well written

1. Tables need to highlight or starred the p values which are significant for a better visualization.

2. Discussion

Discussion needs to reflect all of the results reported in Table 1-3 and studies form the literature should be discussed supporting or not supporting these findings. The discussion should be expanded considering the suggestion above. Currently, it is confusing that if they are discussing their own results or findings from the literature. There area also several statements in which citations are missing.

Some detail points:

1. Second paragraph line 201, that sentence needs to have more than 1 citation. The authors should show more studies which is indication of their expertise in that domain.

2. Second paragraph line 205, the literature demonstrates.... There is no citations added to this sentences. There should be citations supporting this statement.

3. The first sentence of third paragraph. I am confused because of citations, I am assuming they report the finding of this study, then the citations should be explained how those studies support findings of this current study.

4. Same comment ( llike number 3) for this sentence " Furthermore, the positive association of PCMH recognition with nine out of the ten eCQMs, demonstrate the importance of PCMH transformation in high clinical quality performance among HCs. [15-17]" Authors need to clarify this point.

5. line 231..previous should start with capital letter.

6. Line 231..Previous research..... needs citations at the end of the statement

Reviewer #3: Dear authors,

While the topic is relevant and important, you need to do some revisions.

Thank you

Abstract

The aim of the study should be similar with the aim in the main text.

Please privde the references fort he alst paragrapg of the introduction.

Page 7 line 131 add mmHg after blood pressure value

Methods

Page 7- the last paragraph- The number of variables should continue with the number of 9 until 12 since you specified twelve variables.

Discussion section

The section should be supported with more up-to-date references. – especially 1st and 2nd paragraph needs more references

6. PLOS authors have the option to publish the peer review history of their article (what does this mean?). If published, this will include your full peer review and any attached files.

Reviewer #1: Yes: Megan E. Branda

Reviewer #2: No

Reviewer #3: No

---

## [Author Response · Author response to Decision Letter 0]

3 Jan 2020

Response to Reviewers 

Editor Comments: 

Editor Comment #1: Statistical analysis has important concerns. Please rewrite this section using the feedback from reviewers.

Response: The Methods section has been revised based upon reviewer’s comments.

Editor Comment #2: Introduction section can be enriched adding some relevant literature.

Response: Additional relevant literature has been added to enrich the Introduction section.

Editor Comment #3: Discussion section should be reorganized and enriched using the findings in the results section.

Response: The Discussion section has been reorganized to further highlight relevant study findings.

Reviewer#1:

Reviewer #1-1: For the statistical analysis seems incomplete. I am assuming you are aggregating the data at the clinic level, which means you are going to have for any outcome a range from 0-100% for a clinic meeting that quality measure, correct? if this is true then you are not doing bivariate (logistic) regression - you are doing linear regression (or you can do a probit model taking into account the N for each clinic).

or if you are modeling at the patient level then you should be doing a mixed model with a random effect at the clinic level.

Response: We have revised the Methods section and Data Acquisition sub-section to clarify that the Uniform Data System (UDS) collect aggregated data at the health center organizational level (i.e. parent organization or network level). In addition, we have revised the Methods section and Data Analysis sub-section to clarify the bivariate analyses and the multiple linear regression analyses conducted. 

Reviewer #1-2: Concerns regarding model selection - removing variables that do not meet significance (which I am assuming you are using cutoff of 0.05) is not an approach that is recommended for model selection. For your sample size you are not hurting on power so why not adjust by all factors of interest if you do not want to do true model selection?

Also, on any regression model you are making assumptions and you do not state you verify them in the statistical section.

Response: We revised the statement in the Methods section and Data Analysis sub-section to clarify clinical quality measures with statistically non-significant associations with MU stage and practice size were not included in the multiple linear regression analyses.

Reviewer #1-3: For table 2, Only MU is addressed but you are interested in association across 3 key variables of which PCMH is not addresses in terms of describing outcomes (practice size in table 1).

Response: We have added a table (i.e. new Table 3: Electronic Clinical Quality Measure (e-CQMs) Performance by Health Center Practice Size) on the bivariate analyses conducted for practice size and clinical quality performance, and updated the methods and result sections accordingly. As stated originally in the Method section, PCMH status was included in our study to build upon previous research that established associations with clinical quality improvement in health centers. The literature cited were studies that Drs. Suma Nair and Alek Sripipatana had previously conducted. 

Reviewer #1-4: Test statistics in table 1 and 2 were not mentioned and should be described in statistical section (what is p-value testing??).

Response: We have revised the Methods section and Data Analysis sub-section to clarify the bivariate analyses conducted and test statistics used in Tables 1 and 2. 

Reviewer #1-5: The term Multivariate is not used appropriately, you are actually doing Multivariable linear regression.

Response: We have revised the Methods section and Data Analysis sub-section to clarify the multiple linear regression analyses conducted. 

Reviewer #1-6: For presenting modeling results you show coefficient and p-value, the results should show how much variation there is about the estimate.

Response: Regression coefficients inform the mean change in the dependent variable of eCQM mean percentage for one unit of change in the independent variable in MU stage, PCMH recognition, and practice size. Thus, the authors have chosen to highlight and report the coefficients and p-value. 

Reviewer #2:

Reviewer #2-1: This is an important topic given that IT optimization (via meaningful use stage 2) is critical or meant to be critical to improve quality of care. This paper explores association of these measures using public data called UDS. Please see my detailed comments section by section below

Response: We sincerely appreciate the reviewer’s perspective on the significance of our study topic.

Reviewer #2-2: Introduction -The paper needs to add findings form studies which reported on HIT implementation (EHR adoption) and their impact of clinical quality outcomes. There is big literature on that which is missing in the introduction

Response: Thank you for this correction. A paragraph detailing the salient aspects of this evidence has been added to the Introduction. 

Reviewer #2-3: Introduction -There are other studies used the same data set published results for similar or different measures. Some of these studies should be reported

Response: We have added the 2018 study by Kranz AM, Dalton S, Damberg C, Timbie JW). In addition, the authors had included the following references in the initial submission that detailed use of the same data set and examined the association of patient centered medical home transformation and clinical quality measures

• Hu R, Shi L, Sripipatana A, Liang H, Sharma R, Nair S, et al. The Association of Patient-centered Medical Home Designation With Quality of Care of HRSA-funded Health Centers: A Longitudinal Analysis of 2012-2015. Medical care. 2018;56(2):130-8.

• Shi L, Lee DC, Chung M, Liang H, Lock D, Sripipatana A. Patient-Centered Medical Home Recognition and Clinical Performance in U.S. Community Health Centers. Health services research. 2017;52(3):984-1004.

• Shi L, Lock DC, Lee DC, Lebrun-Harris LA, Chin MH, Chidambaran P, et al. Patient-centered Medical Home capability and clinical performance in HRSA-supported health centers. Medical care. 2015;53(5):389-95.

Reviewer #2-4: Methods - The method section should be formatted well with sub headings such as study design, data Acquisition, data analysis, validity etc.

Response: We have revised to add subheadings of data acquisition, study design, and data analysis to the Results section.

Reviewer #2-5: Methods - They need to add hypothesis and state if hypothesis are supported or not.

Response: We have revised the Methods section and Study Design sub-section with the hypothesis. In addition, we have revised the specific aim statement in the Introduction section to add further clarity on the objective of the study.

Reviewer #2-5: Results - It is straightforward and well written

Response: We sincerely appreciate the reviewer’s comment.

Reviewer #2-6: Results -Tables need to highlight or starred the p values which are significant for a better visualization.

Response: We have revised the tables by bolding the p-values to highlight the statistically significant findings.

Reviewer #2-7: Discussion - Discussion needs to reflect all of the results reported in Table 1-3 and studies form the literature should be discussed supporting or not supporting these findings. The discussion should be expanded considering the suggestion above. Currently, it is confusing that if they are discussing their own results or findings from the literature. There area also several statements in which citations are missing.

Response: We have revised the language used in this section to simplify and to clarify the focus of Discussion. 

Reviewer #2-8: Second paragraph line 201, that sentence needs to have more than 1 citation. The authors should show more studies which is indication of their expertise in that domain.

Response: Citations have been added to strengthen this statement. 

Reviewer #2-9: Second paragraph line 205, the literature demonstrates.... There is no citations added to this sentences. There should be citations supporting this statement.

Response: Citations have been added.

Reviewer #2-10: The first sentence of third paragraph. I am confused because of citations, I am assuming they report the finding of this study, then the citations should be explained how those studies support findings of this current study.

Response: This has been clarified to reflect where this study stands in comparison to other studies. 

Reviewer #2-11: Same comment ( like number 3) for this sentence " Furthermore, the positive association of PCMH recognition with nine out of the ten eCQMs, demonstrate the importance of PCMH transformation in high clinical quality performance among HCs. [15-17]" Authors need to clarify this point.

Response: We have removed the citations in order to clarify that this statement refers to results from the study. 

Reviewer #2-12: line 231..previous should start with capital letter.

Response: This change has been made. 

Reviewer #2-12: Line 231..Previous research..... needs citations at the end of the statement

Response: Citations have been added.

Reviewer #3: 

Reviewer #3-1: Abstract - The aim of the study should be similar with the aim in the main text.

Response: Text has been revised on the aims of the study. 

Reviewer #3-2: Please provide the references for the last paragraph of the introduction.

Response: A reference has been added.

Reviewer #3-3: Page 7 line 131 add mmHg after blood pressure value

Response: This has been added to the text. 

Reviewer #3-4: Methods - Page 7- the last paragraph- The number of variables should continue with the number of 9 until 12 since you specified twelve variables.

Response: We have revised the text in the paragraph to indicate the 8 preventive care eCQMs and 4 chronic care eCQMs collected. 

Reviewer #3-4: Discussion section -The section should be supported with more up-to-date references. – especially 1st and 2nd paragraph needs more references

Response: Additional references have been incorporated to the Discussion section.

---

## [Decision Letter · Decision Letter 1]

6 Feb 2020

PONE-D-19-29335R1

Impact of Health Information Technology Optimization on Clinical Quality Performance in Health Centers: A National Cross-Sectional Study

PLOS ONE

Dear Dr. Lin,

Thank you for submitting your manuscript to PLOS ONE. After careful consideration, we feel that it has merit but does not fully meet PLOS ONE’s publication criteria as it currently stands. Therefore, we invite you to submit a revised version of the manuscript that addresses the points raised during the review process.

I can not recommend acceptance until the statistics part is free from major flaws as highlighted by one of the reviewers. Please consider getting statistics consultancy.

We would appreciate receiving your revised manuscript by Mar 22 2020 11:59PM. To enhance the reproducibility of your results, we recommend that if applicable you deposit your laboratory protocols in protocols.io, where a protocol can be assigned its own identifier (DOI) such that it can be cited independently in the future. For instructions see: http://journals.plos.org/plosone/s/submission-guidelines#loc-laboratory-protocols

We look forward to receiving your revised manuscript.

Kind regards,

Mustafa Ozkaynak

Academic Editor

PLOS ONE

Additional Editor Comments (if provided):

Although revisions have significantly improved the manuscript, statistical part is still concerning. Authors are encouraged to pay attention to reviewers' comment and address the issues they (particularly one of them) raised. I highlight recommend getting consultancy from a statistician since the manuscript includes major statical flaws that decrease the credibility of the study and could be caught by a statistician.

Reviewers' comments:

Reviewer's Responses to Questions

**Comments to the Author**

1. If the authors have adequately addressed your comments raised in a previous round of review and you feel that this manuscript is now acceptable for publication, you may indicate that here to bypass the “Comments to the Author” section, enter your conflict of interest statement in the “Confidential to Editor” section, and submit your "Accept" recommendation.

Reviewer #1: (No Response)

Reviewer #3: All comments have been addressed

2. Is the manuscript technically sound, and do the data support the conclusions?

Reviewer #1: Partly

Reviewer #3: Yes

3. Has the statistical analysis been performed appropriately and rigorously? 

Reviewer #1: No

Reviewer #3: Yes

4. Have the authors made all data underlying the findings in their manuscript fully available?

Reviewer #1: No

Reviewer #3: Yes

5. Is the manuscript presented in an intelligible fashion and written in standard English?

Reviewer #1: Yes

Reviewer #3: Yes

6. Review Comments to the Author

Reviewer #1: First I want to thank the authors for being responsive to my previous review.

I still have some minor critiques to the statistical analysis that need to be addressed:

* on line 132 and in several other places you call the analysis bivariate. This is incorrect. Bivariate is a two level analysis, when you reference it is in terms of either the HC size (small, medium, large) or just being added without actual context. This should be removed from all text.

* within the results you list Statistical Significance, There has been a big shift in the literature to avoid this term and the ASA recommends to not use this term. Instead we should be thinking of clinical relevance. When reading your results I am not sure how much of the p-value is driven by sample size. Which a lot of that comes from presenting the information to allow the reader to interpret the findings accurately beyond your text.

When reporting the model results you should be showing the Coefficient, the SE (Standard Error), 95% CI, Test statistic and p-value. Each of your models should report the overall model effect as well the R^2 associated with each. Also, it is typical to see all coefficients including intercept. You are trying to show a lot of information in the table so I can see the justification in only concentrating on the three variables of interest. In your response to showing SE, this is not acceptable. The reader needs to understand the variation about that change.

Also, you do not mention model assumptions, an appropriate model selection was not done and you do not mention interaction testing.

Reviewer #3: All revision request are appropriate within the manuscript regarding my concerns. Thank you for your effort!

7. PLOS authors have the option to publish the peer review history of their article (what does this mean?). If published, this will include your full peer review and any attached files.

Reviewer #1: Yes: Megan E. Branda

Reviewer #3: No

---

## [Author Response · Author response to Decision Letter 1]

22 Mar 2020

Response to Reviewer #1:

Reviewer #1-1 had stated a “No” response to Q4: Have the authors made all data underlying findings in their manuscript fully available? 

Response: The authors had previously provided information in the manuscript submission questionnaire on how to request access to the Uniform Data System as follows: The Health Resources and Services Administration/Bureau of Primary Health Care (HRSA/BPHC)'s Health Center Program makes the Uniform Data Systems (UDS) available, for free, to the public in an electronic format. This action is being taken pursuant to the Freedom of Information Act (5 U.S.C. 552(a)(2)(D)) (FOIA). URL: https://www.hrsa.gov/foia/UDS-public-use.html) 

Reviewer #1-2: On line 132 and in several other places you call the analysis bivariate. This is incorrect. Bivariate is a two level analysis, when you reference it is in terms of either the HC size (small, medium, large) or just being added without actual context. This should be removed from all text.

Response: Bivariate analysis is the analysis of two variables to study the empirical relationship between the variables.[1] Examples of the types of variable of interest could be binary, categorical, ordinal, nominal, and continuous. In Table 1, we analyzed practice size and each of the characteristics in the bivariate analysis. Practice size is a categorical variable with three categories of small, medium, and large. 

Reviewer #1-3: within the results you list Statistical Significance, There has been a big shift in the literature to avoid this term and the ASA recommends to not use this term. Instead we should be thinking of clinical relevance. When reading your results I am not sure how much of the p-value is driven by sample size. Which a lot of that comes from presenting the information to allow the reader to interpret the findings accurately beyond your text.

Response: P-value is one of the tools in statistical hypothesis testing. Thus, we have chosen to report it in our manuscript. In response the Reviewer #1 comment 1-4, we have revised Table 4 to report the 95% confidence interval along with p-values to provide the reader with additional statistics in interpreting the findings. 

With respect to the comment on clinical relevance, the study conducted a data analysis of the Uniform Data System (UDS), which is an administrative data set containing information on patient sociodemographic characteristics, primary care services provision, healthcare workforce, clinical quality measures (CQMs), and Meaningful Use (MU) attestation that are reported and aggregated at the health center organizational level as described in the Methods section. This study aims to examine the association between MU attainment supported by federal initiative and clinical quality performance at the health center organization level as stated in the Introduction section. This was not a clinical treatment study. No individual patient records are contained in UDS dataset. Thus, clinical relevance does not pertain to our study. 

Reviewer #1-4: When reporting the model results you should be showing the Coefficient, the SE (Standard Error), 95% CI, Test statistic and p-value. Each of your models should report the overall model effect as well the R^2 associated with each. Also, it is typical to see all coefficients including intercept. You are trying to show a lot of information in the table so I can see the justification in only concentrating on the three variables of interest. In your response to showing SE, this is not acceptable. The reader needs to understand the variation about that change.

Response: Table 4 has been updated with coefficient, standard error, 95 CI, test statistic and p-value. 

Reviewer #1-5: Also, you do not mention model assumptions, an appropriate model selection was not done and you do not mention interaction testing.

Response: We had conducted interaction testing, which demonstrated that interacting terms were not statistically significant. Thus, interaction terms were not retained in the regression model. [2] We have added a statement in the Methods section and reference in the revised manuscript. 

References 

1. Tabachnick BG, Fidell LS. Using multivariate statistics (5th ed). Boston, MA: Allyn & Bacon/Pearson Education; 2007.

2. Pagano M, Gauvreau K. Principles of Biostatistics 2nd Edition. Australia: Duxbury; 2000.

---

## [Decision Letter · Decision Letter 2]

20 May 2020

PONE-D-19-29335R2

Impact of health information technology optimization on clinical quality performance in health centers: A national cross-sectional study

PLOS ONE

Dear Dr. Lin,

Thank you for submitting your manuscript to PLOS ONE. After careful consideration, we feel that it has merit but does not fully meet PLOS ONE’s publication criteria as it currently stands. Therefore, we invite you to submit a revised version of the manuscript that addresses the points raised during the review process.

We would appreciate receiving your revised manuscript by Jul 04 2020 11:59PM. To enhance the reproducibility of your results, we recommend that if applicable you deposit your laboratory protocols in protocols.io, where a protocol can be assigned its own identifier (DOI) such that it can be cited independently in the future. For instructions see: http://journals.plos.org/plosone/s/submission-guidelines#loc-laboratory-protocols

We look forward to receiving your revised manuscript.

Kind regards,

Mustafa Ozkaynak

Academic Editor

PLOS ONE

Additional Editor Comments (if provided):

Please address the issue highlighted by one of the reviewers about percentages.

Reviewers' comments:

Reviewer's Responses to Questions

**Comments to the Author**

1. If the authors have adequately addressed your comments raised in a previous round of review and you feel that this manuscript is now acceptable for publication, you may indicate that here to bypass the “Comments to the Author” section, enter your conflict of interest statement in the “Confidential to Editor” section, and submit your "Accept" recommendation.

Reviewer #4: (No Response)

Reviewer #5: (No Response)

2. Is the manuscript technically sound, and do the data support the conclusions?

Reviewer #4: Yes

Reviewer #5: Yes

3. Has the statistical analysis been performed appropriately and rigorously? 

Reviewer #4: Yes

Reviewer #5: No

4. Have the authors made all data underlying the findings in their manuscript fully available?

Reviewer #4: Yes

Reviewer #5: Yes

5. Is the manuscript presented in an intelligible fashion and written in standard English?

Reviewer #4: Yes

Reviewer #5: Yes

6. Review Comments to the Author

Reviewer #4: The authors should provide a more detailed explanation of the following

The point should be placed at the end of the reference.

Reviewer #5: I think more clarification is required as to why mean percentages is used in the analysis. By using the mean percentages, some of the categories, for example, small practice size may have varying responses for cervical cancer screening. Any outliers, or differing ranges will not be reflected if the mean percentages are used, and it may affect the analysis that should be undertaken. If the authors can provide more justification around the use of mean percentages it will help determine whether the analysis is appropriate and provides more context to the data.

7. PLOS authors have the option to publish the peer review history of their article (what does this mean?). If published, this will include your full peer review and any attached files.

Reviewer #4: No

Reviewer #5: No

---

## [Author Response · Author response to Decision Letter 2]

22 Jun 2020

Response to Reviewers 

Reviewer #4: The authors should provide a more detailed explanation of the following

The point should be placed at the end of the reference.

Response: Authors have reviewed all reference citations and revised the manuscript to place the period to the end of citation. The authors found a paper published in PLOS ONE in June of 2020 entitled “Echocardiographic screening to determine progression of latent rheumatic heart disease in endemic areas: A systematic review and meta-analysis” that followed the citation format of our manuscript (https://doi.org/10.1371/journal.pone.0234196); on the hand, there was another paper entitled “’It's disappointing and it's pretty frustrating, because it feels like it's something that will never go away.’ A qualitative study exploring individuals’ beliefs and experiences of Achilles tendinopathy” published in Mary of 2020 with the period placed at the end of reference (https://doi.org/10.1371/journal.pone.0233459). Thus, we have included our suggestion in the editor’s letter to update the online submission guidelines with citation format examples that may address potential confusion in the future. 

Reviewer #5: I think more clarification is required as to why mean percentages is used in the analysis. By using the mean percentages, some of the categories, for example, small practice size may have varying responses for cervical cancer screening. Any outliers, or differing ranges will not be reflected if the mean percentages are used, and it may affect the analysis that should be undertaken. If the authors can provide more justification around the use of mean percentages it will help determine whether the analysis is appropriate and provides more context to the data.

Response: We have added a statement in the Methods section on the use of mean percentages as the most commonly used measure of central tendency and citation in the revised manuscript. With regards to the outlier concerns, the justification is due to the nature of the dataset. Our study conducted data analysis on the Uniform Data System (UDS), which is an administrative data set that is reported and aggregated at the health center organizational level for all patients served as described in the Methods section. Health centers must report UDS annually as a requirement of receiving federal grant funding. Thus, all clinical quality measurements reported, including outliers, are legitimate observations that are part of the health center organization reporting. Hence, we have included data from all health center organizations in the analysis.

---

## [Editor Report · Decision Letter 3]

29 Jun 2020

Impact of health information technology optimization on clinical quality performance in health centers: A national cross-sectional study

PONE-D-19-29335R3

Dear Dr. Lin,

We’re pleased to inform you that your manuscript has been judged scientifically suitable for publication and will be formally accepted for publication once it meets all outstanding technical requirements.

Kind regards,

Mustafa Ozkaynak

Academic Editor

PLOS ONE
---

## [Editor Report · Acceptance letter]

6 Jul 2020

PONE-D-19-29335R3 

Impact of health information technology optimization on clinical quality performance in health centers: A national cross-sectional study 

Dear Dr. Lin:

I'm pleased to inform you that your manuscript has been deemed suitable for publication in PLOS ONE. Congratulations! Your manuscript is now with our production department. 

Kind regards, 

on behalf of

Dr. Mustafa Ozkaynak 

Academic Editor

PLOS ONE